# Why do GPs leave direct patient care and what might help to retain them? A qualitative study of GPs in South West England

Anna Sansom,[1] Rohini Terry,[2] Emily Fletcher,[1] Chris Salisbury,[3] Linda Long,[2] Suzanne H Richards,[4] Alex Aylward,[5] Jo Welsman,[6] Laura Sims,[1] John L Campbell,[1] Sarah G Dean[2]

¹Primary Care Research Group, University of Exeter Medical School, Exeter, UK
²University of Exeter Medical School, College House, Exeter, UK
³Centre for Academic Primary Care, Population Health Sciences, Bristol Medical School, University of Bristol, Bristol, UK
⁴Academic Unit of Primary Care, Leeds Institute of Health Research, University of Leeds, Leeds, UK
⁵Patient and Public Involvement Group, NIHR CLAHRC, Exeter, UK
⁶Centre for Biomedical Modelling and Analysis, Living Systems Institute, University of Exeter, Exeter, UK

**Correspondence to**
Professor John L Campbell;
john.campbell@exeter.ac.uk

## ABSTRACT

**Objective** To identify factors influencing general practitioners' (GPs') decisions about whether or not to remain in direct patient care in general practice and what might help to retain them in that role.

**Design** Qualitative, in-depth, individual interviews exploring factors related to GPs leaving, remaining in and returning to direct patient care.

**Setting** South West England, UK.

**Participants** 41 GPs: 7 retired; 8 intending to take early retirement; 11 who were on or intending to take a career break; 9 aged under 50 years who had left or were intending to leave direct patient care and 6 who were not intending to leave or to take a career break. Plus 19 stakeholders from a range of primary care-related professional organisations and roles.

**Results** Reasons for leaving direct patient care were complex and based on a range of job-related and individual factors. Three key themes underpinned the interviewed GPs' thinking and rationale: issues relating to their personal and professional identity and the perceived value of general practice-based care within the healthcare system; concerns regarding fear and risk, for example, in respect of medical litigation and managing administrative challenges within the context of increasingly complex care pathways and environments; and issues around choice and volition in respect of personal social, financial, domestic and professional considerations. These themes provide increased understanding of the lived experiences of working in today's National Health Service for this group of GPs.

**Conclusion** Future policies and strategies aimed at retaining GPs in direct patient care should clarify the role and expectations of general practice and align with GPs' perception of their own roles and identity; demonstrate to GPs that they are valued and listened to in planning delivery of the UK healthcare; target GPs' concerns regarding fear and risk, seeking to reduce these to manageable levels and give GPs viable options to support them to remain in direct patient care.

## INTRODUCTION

General practitioners (GPs) make numerous, complex decisions about patient care and

### Strengths and limitations of this study

► A maximum variation general practitioner (GP) sample was achieved through use of the workforce census survey returns.
► The large number of interviews conducted provided rich data and the opportunity to explore opinions and experiences with GPs and stakeholders.
► The analysis process was supported through study input from the Patient and Public Involvement group and a GP representative.
► GPs were self-selecting and represented mainly those indicating the intention to leave (or who had already left) direct patient care.

service delivery on a daily basis. Whether or not to remain in general practice and whether that should involve a direct patient care role, are additional decisions many GPs are facing at a time when the GP workforce is said to be in 'crisis'.[1–3] Our recent survey of all GPs in South West England revealed that 37% of GPs reported a high likelihood of quitting direct patient care within 5 years and that, overall, 70% of GPs reported a career intention that, if implemented, would negatively impact GP workforce capacity and availability in the next 5 years.[4] Similar figures have been found elsewhere in the UK[5–8] and, along with an overall reduction in the number of GPs, unfilled training places and an ageing workforce,[9] represent a toxic challenge to primary care provision in the UK.

Our systematic review of survey-based studies of the UK GPs identified four job-related factors that contributed to GPs' decision-making about leaving their jobs or reducing their hours: workload, job (dis)satisfaction, work-related stress and work-life balance.[10] However, many other detailed factors underlie these job-related factors or

**Table 1** Number of general practitioner (GP) interviewees in each interview category (n=41)

| GP interview category | Number interviewed |
|---|---|
| Retired GPs (age 50+ years) | 7 |
| GPs intending to retire (50–60 years) | 8 |
| GPs on or intending to take a career break (any age) | 11 |
| GPs who had quit or were intending to quit (35–49 years) | 9 |
| Staying GPs (any age; 'high' or 'very high' morale) | 6 |
| Total | 41 |

may influence decisions.[10] Few qualitative studies provide a contemporary view of decision-making or include GPs of more than one age group or decision outcome.[11] There is also a lack of evidence to support the view that recent policies and strategies developed to address GP workforce retention (eg, as outlined in the General Practice Forward View[12]) will help to reverse the GP shortage. In the meantime, existing GPs of all ages still face the decision about whether or not to remain in direct patient care. The aim of this study was to gain insight into the lived experiences of GPs in today's National Health Service (NHS), with the intention of identifying factors which, if addressed, might facilitate retention in direct patient care. This was part of a larger mixed methods study seeking to inform the development of policies and strategies to support the retention of experienced GPs (ReGROUP).

## METHOD

During the course of a survey of all GPs in South West England,[4] GPs were asked about their future intentions (within the next 5 years) regarding remaining in, leaving or taking a break from direct patient care, or reducing their hours. A sample population was drawn from survey respondents who reported being willing to be interviewed and who met the eligibility criteria for one of five interview categories (table 1). A maximum variation approach was used to identify a purposive sample of GPs from practices of varying size and deprivation, GP demographic profiles and GP role (partner, salaried or locum).

We created a second sampling frame of key stakeholders to obtain the views of other professionals with direct experience of GP workforce issues and who could comment on the impact on practice organisation and management. Convenience sampling was used to identify individuals from a range of healthcare roles and professional organisations to be approached for interview within South West England.

An invitation letter, information sheet and consent form were sent to each potential participant. A maximum of three attempts were made to contact and schedule an interview before moving on to the next potential

participant in the sampling frame. We paused halfway through recruitment to review the sample and to determine whether adjustments were needed in our sampling approach.

Semi-structured interview schedules were developed using themes identified from the literature and through discussion with the study's patient and public involvement (PPI) members and GP representative. Individual interviews were conducted face-to-face, by telephone or by Skype video call depending on the participant's preference. Interviews were conducted by one of two experienced, postdoctoral, female, qualitative researchers (AS and RT) from May to November 2016. Interviews were audio-recorded, transcribed verbatim and anonymised. Interviewees provided consent prior to interview and were offered a gift voucher in acknowledgement of their time.

Transcripts from GP and stakeholder interviews were analysed together. Transcribed interviews were entered into data management software QSR NVivo V.11[13] and analysed using thematic analysis. An initial coding frame was independently constructed by AS and RT, based on five transcripts. Following discussions, a consensus about the coding frame was reached. The coding frame was then independently tested by AS and RT with two further interview transcripts and final modifications were made. All transcripts were coded using this final coding frame. Detailed project notes were kept regarding further refinement of any existing, or the addition of new, codes. To aid trustworthiness and reduce any potential bias, the researchers wrote field notes and reflexive memos and discussed these during peer debriefing sessions.

Discussions to help analyse emerging themes were held by the research team (AS, RT and SD) along with the PPI group and a GP representative (who was not a study participant). Recruitment and analysis were concurrent; data collection concluded when code and meaning saturation had been reached.[14] A written summary of the findings was shared with all participants at the end of the study.

### Patient and public involvement

Patient representatives were recruited from local PPI networks and chaired by JW, Engaged Research Fellow. The group consisted of two men and five women representing individuals with experience of a range of long-term physical and mental health conditions. Some of the group also had experience as carers for elderly relatives or for children with life-altering health conditions. One member had extensive experience as a lay representative for Clinical Commissioning Groups and Quality Outcome Framework as a lay assessor of GP practices.

PPI members were involved throughout this study, using their experiences and patient perspectives to contribute to several key stages including: initial study design and funding application; review of application for ethical approval and development of qualitative interview questions. They also took part in group discussions with the

| Table 2 Demographic distribution of GP sample (n=41) | |
|---|---|
| GP role (current or most recent) | |
| Partner | 22 |
| Salaried | 5 |
| Locum | 14 |
| Gender | |
| Female | 21 |
| Male | 20 |
| Ethnicity | |
| White | 41 |
| Other | 0 |
| Age (years) | |
| <40 | 7 |
| 40–49 | 16 |
| 50+ | 18 |
| Practice list size* | |
| <3500–8000 | 8 |
| >8000 | 21 |
| Not known† | 12 |
| Practice Index of Multiple Deprivation (IMD 2015) score* | |
| 1–5 (more deprived) | 10 |
| 6–10 (less deprived) | 19 |
| Not known* | 12 |

*Public Health England: http://fingertips.phe.org.uk/profile/general-practice
†GPs who were locums and attached to more than one practice.
GP, general practitioner.

| Table 3 Stakeholder roles and organisations (n=19) | |
|---|---|
| Role/organisations represented | Number interviewed |
| Clinical Commissioning Group | 4 |
| Local Medical Committee | 3 |
| Care Quality Commission | 2 |
| Regional organisation | 1 |
| General practitioner with interest in workforce issues | 4 |
| Practice manager | 1 |
| Nurse practitioner | 1 |
| Other allied health professional | 1 |
| Pharmacist | 2 |
| Total | 19 |

researchers to help refine themes and interpret the data during the analysis process; and contributed to discussion about the implications of the findings. A final dissemination and debriefing workshop took place with the PPI members at the end of the study.

## RESULTS

Thirty-one per cent (694/2248) of those who returned the GP workforce census survey were eligible for interview. Invitations were sent to a purposive sample of 98 GPs and to a convenience sample of 41 stakeholders; 44 GPs and 19 stakeholders agreed to be interviewed. Interviews were conducted by telephone (31 GPs; 15 stakeholders), face-to-face (8 GPs; 4 stakeholders) and Skype (2 GPs). Interviews lasted between 15 and 73 min. Scheduling difficulties resulted in three GPs who had agreed to interview not being interviewed.

Interviewees were recruited from urban and rural areas across South West England. GP interviewees included partner, salaried and locum GPs. Table 1 reports the number of GPs interviewed in each interview category. Table 2 illustrates participants' demographic and practice characteristics (where known). Table 3 reports stakeholders' roles and the organisations they represented.

Stakeholders provided greater perspectives of the broader issues; 11 of the stakeholders had also been, or were currently, GPs (in addition to any other role).

In-depth analysis of interview data revealed three themes underpinning GPs' thinking and rationale in respect of continuing to provide direct patient care: issues relating to their personal and professional identity and the perceived value of general practice-based care within the healthcare system; concerns regarding fear and risk, for example, in respect of medical litigation and managing administrative challenges within the context of increasingly complex care pathways and environments; and issues around choice and volition in respect of personal social, financial, domestic and professional considerations. These themes are presented in this paper. Additional findings from the interviews, broader than the scope of this paper, are provided in our full ReGROUP study report to the funder (NIHR study number 14/196/02).

### Identity and value

Three subthemes related to identity and value were identified.

#### Boundaries and expectations for general practice

Changes within general practice have led to diminished clarity around professional boundaries and unrealistic expectations about what general practice can (and should) deliver. Tensions were identified between the primary and secondary care interface: general practice tended to 'pick up' and manage aspects of care that were felt to be the remit of other services and there was a clear sense that the 'buck stops' with general practice:

> GPs, being the out of hospital doctors, have had to pick up everything (…) And eventually the buck stops with the GP. (GP stakeholder)

#### GP identity, professionalism and morale

Identity and professionalism were important to GPs:

…being a doctor, for me and I think for most of my colleagues is about professionalism (…) Don't need a carrot or a stick – just love the job. Just want to get on with it (…) they will miss us when we're gone. (GP partner, male, aged 50–59 years, retired)

However, many felt compromised in their ability to practise in their preferred ways (eg, in the length of time they were able to spend with patients) and this impacted negatively on morale, professionalism and identity. This was partly due to: unrealistic demands; concerns about complaints; targets and guidelines; complexity of cases and lack of time to address them; lack of continuity and loss of professional autonomy. GPs felt compromised in striking a balance between delivering high-quality care and doing this within the constraints and burdens they experienced:

I like to listen to people, I like to give people time, so I'm a very popular GP, but I'm quite a stressed GP in the NHS setting (GP locum, female, aged 30–39 years, intending career break)

…to survive in today's NHS you have to be comfortable taking risks and cutting corners (CCG stakeholder)

### Being listened to and being valued

GPs expressed frustration over not feeling listened to. They felt strongly that the government had failed to listen to them, to general practice as a profession and to the British Medical Association, about the impending workforce crisis. This was mirrored in matters relating to organisational change and demands on the service.

Feeling valued on account of their work was important to GPs. However, they reported often feeling undervalued by the general public, the NHS, the media and the government:

I think most people, if you ask them why they do jobs, it's a complex mixture and a lot of it comes about being valued and appreciated. I mean, people always focus on incomes and things but, the more detailed the analysis is, it always comes back to things like being appreciated, feeling valued (GP partner, male, age 60+ years, intending retirement)

### Fear and risk

Fear and anxiety were experienced regarding different aspects of risk that had to be managed within the GP role. There was a general perception that GPs are good at managing clinical risk. However, risks were perceived to have increased in recent years, practice had changed to accommodate them and the risks were "not proportional to the rewards" (GP partner, male, age 30–39 years, intending career break). Three subthemes were identified.

### Risk to patient care and safety and fear of complaints and being sued

There were concerns about the safety of practice and the quality of care being delivered to patients:

…when repeat prescriptions came through to be re-authorised I would be checking through and making sure everything was up to date and everybody else (…) was just re-authorising it because they'd given up that aspect (…) of safety… (GP locum, female, aged 40–49 years, staying)

Risk was related to 'unmanageable' workloads, the complexity of cases, the large number of decisions that had to be made and the impact of cumulative decision-making throughout the day:

…you have to balance priorities and triage things and I think (…) the busier you get the more dangerous your decision-making becomes on that front and the riskier it can get (GP partner, female, aged 40–49 years, staying)

Fear of making mistakes and litigation influenced how medicine was being practised, with some GPs practising more defensively (eg, spending more time writing notes and choosing face-to-face rather than telephone consultations):

We don't really practice evidence-based medicine; we practice a sort of legal-based medicine (GP stakeholder)

Patients were perceived to have easy routes to complain about their GP/practice, but there was felt to be little support or recourse for GPs in actively managing those concerns. Defensive practice was also seen as a response to patient expectations with one GP feeling that "I'd better give this person what they want or they will complain" (salaried GP, male, aged 40–49 years, intending early leaver).

Where GPs had experienced complaints or being sued, they described the process as being drawn out and stressful. Complaints "wound the doctor severely (…). When you're kicked in the teeth like that, either by the government or the patient, it really hurts" (GP stakeholder).

### Risk to professional status and identity and to own health and well-being

Participants described GPs who they felt 'cut corners' and other GPs who did not. Both routes could potentially create a risk and be a threat to the professional status of the GP, their well-being and morale and the profession overall:

GPs tend go down one of two routes: they either- to cope with demand - start to cut corners (…) or (…) you over burden yourself and you won't cut corners (…) and that has its consequences at the end of the day (locum GP, male, aged 40–49 years, intending career break)

The consequences of experiencing ongoing fear and anxiety and of having to manage a range of risks, impacted negatively on GPs' own health and well-being.

Work pressures had led to GP colleagues taking sick leave; some participants had direct experience of their own work-related ill health:

> I was just working at such a pace and I knew I was making myself ill (locum GP, female, aged 50–59 years, intending retirement)

The fear of becoming ill compounded the fear of making mistakes, creating a vicious circle:

> You can't make yourself ill. If you make yourself ill, you're going to make mistakes anyway and no one wants that (GP partner, male, aged 30–39 years, intending career break)

### Financial risk and uncertainty about the future of general practice

Financial investment in a practice was perceived to be a greater risk now than in previous times and this was both a burden of and a barrier to, investment. Buying into a practice could mean taking on the risk of personal debt and increased stress. GPs may have previously been willing to make a long-term financial investment in general practice, but other pressures on personal finances, uncertainty about the viability of long-term commitment and concerns about the future of general practice, meant that younger GPs (in particular) were now reluctant to invest:

> …if I had been willing to take on the whole practice and just tough it out, there's a chance that in 20 years I would have £800 000 of equity in a building, but there is an equal chance I would burn out, be reported to the GMC, gone crazy… (GP partner, male, aged 30–39 years, intending career break)

Older GPs who had invested were experiencing stress and anxiety due to concerns about changes to practice mortgages, the threat of having to make staff redundant or practice closure and responsibilities arising from joint civil liability for a practice.

There was a pessimistic view of the future of the NHS and general practice:

> There seems to be a lack of belief that the NHS will survive, let alone GPs part of it (CQC stakeholder)

Pessimism and uncertainty directly impacted on decisions about staying in or returning to direct patient care following a career break. One GP, on a career break at the time of the interview, saw the current workforce 'crisis' as a barrier to returning to practice:

> …it feels like something in crisis and who wants to jump into that? (GP partner, male, aged 50–59 years, on career break)

There was frustration over a perceived lack of ability to determine the future of general practice and the lack of a unified model that could be implemented:

> There is so much uncertainty and the biggest frustration of being a GP is that you're beholden to whatever the NHS England decision is, or whatever the Department of Health's decision is… (GP partner, male, aged 40–49 years, staying)

### Choice and volition

This theme concerned GPs' feelings about making their decisions to leave or to remain in direct patient care and the degree of choice they felt they actually had. Three subthemes were identified.

### Cumulation, compounding and combination of factors; decisions do not happen in isolation

A range of inter-related factors contributed to GPs' decision-making: factors relating to workload, their practice, their personal circumstances and the wider social context (eg, 'GP bashing' by the media). The accumulation and compounding of factors over a number of years could ultimately lead to decisions to leave or to reduce hours:

> It's really like an insidious, drip drip drip thing really that's been happening for ten plus years, really. There's more and more and more things coming our way. (GP locum, male, aged 50–59 years, intending retirement)

For some GPs there had been a key tipping point for their decision making:

> …everything happened at once: the menopause, the awful complaint (…) the locum work that I wasn't particularly enjoying (…) and I got to the stage of thinking, 'I don't have to do this. I'm not enjoying it. Why am I doing it? Let's just stop and see if I miss it.' (GP locum, female, aged 50–59 years, retired)

Individual career decisions could affect colleagues, peers, patients and the profession in general and this was taken into consideration by the GPs. For example, retiring early from a partner position when the practice was experiencing recruitment difficulties, or choosing to work part-time knowing that others would need to provide cover:

> And if individual partners jumped ship, it was incredibly disruptive (…) Certainly, that had a knock-on effect, not just within the doctors who are the partners, but the wider staff, the nurses, the receptionists, everybody. And it was a less good place to come to work. (GP partner, male, aged 50–59 years, retired)

GPs were also mindful that a decision to remain could have a negative impact on others in the practice:

> The worry is about being miserable around people who don't need misery (…) Like I say, sever the gangrenous limb and you save the patient! (…) the best thing you could do is leave so that actually you're not polluting in any way. (GP partner, male, aged 50–59 years, intending retirement)

### GP resilience and the only route left

Interviewees felt that GPs' resilience had been eroded over recent years. This erosion was linked to loss of control:

> …not feeling in control of where the money's coming from, not feeling in control of your future because if you're going to have contracts imposed on you by the government, you're not in control. So that's where I feel the loss of resilience is coming from… (Salaried GP, female, aged 40–49 years, staying)

A number of participants felt strongly that the solution to the current workforce crisis was not simply to make GPs more resilient but rather that the system they work in needs to be addressed:

> If the purpose of resilience is to enable the same workforce to cope with every increasing demand, that's not on, we actually have to make the job doable. (GP stakeholder)

> …these are people who are highly resilient already (…) the system is so cruel (…). You've got to make changes to the system. Just supporting people is the wrong approach. (GP locum, female, aged 40–49 years, early leaver)

It was noted that stigma exists around GPs accessing help—particularly mental health support—and GPs expressed concerns about confidentiality. Concern was also highlighted over GPs exhausting all alternative routes and coping strategies (such as changing to part-time or portfolio working) and feeling that the only option they had left was to 'vote with their feet' and leave direct patient care:

> I've just become more and more desperate (…) in past years I have just felt terribly angry with the way things are going and now I think, 'I can't actually do anything more about it'. And if I could do anything but vote with my feet, but ultimately it's the only vote which they're going to listen to. (GP partner, male, aged 50–59 years, intending retirement)

### DISCUSSION

Recent surveys suggest that approximately one in three GPs intend to leave direct patient care within 5 years.[4–8] Our findings paint a complex and bleak picture of GPs' experiences and illustrate underlying factors that may be contributing to the large number of GPs leaving or considering leaving direct patient care. The Government has identified the need for an additional 5000 GPs by 2020[12] and retaining existing GPs in direct patient care is a critical issue because of the time lag before newly trained GPs start to practice. Concern has been raised that 'if general practice fails, the whole NHS fails'.[3] Thus, there is an urgent need to better understand GPs' experiences and decision-making rationales to inform any policies and strategies aimed at retaining them and also to contextualise any evaluations of the effects and impacts of new and existing policies and strategies.

Our previous pilot research with older GPs (aged 50–60 years) identified four key themes that highlighted individual and job-related factors associated with decisions about remaining in general practice: early retirement is a viable option for many GPs; GPs have employment options other than undertaking direct patient care; GPs report feeling they are doing an (almost) undoable job and GPs may have other aspirations that pull them away from direct patient care.[15] A study of younger GPs (aged <50 years) who had left direct patient care identified the changing role of general practice as key (including: organisational changes; clash of values; increased workload; negative media portrayal; workplace issues and lack of support).[16] Our synthesis of qualitative studies identified three central dynamics key to understanding the UK GP quitting behaviour: factors associated with low job satisfaction, high job satisfaction and those linked to the doctor-patient relationship.[17] The current study explored how job-related and individual factors are experienced by GPs—their 'lived experience'. We sampled from a broad range of GPs and other primary care stakeholders and identified three central themes which underpin decision-making: identity and value, fear and risk and choice and volition.

Workplace theories and models provide insight into the significance of these findings. The Theory of Organisational Justice highlights the importance of feeling valued and treated fairly in the workplace.[18] GPs in our current study repeatedly described a perception of unfairness and feeling undervalued (often using colloquialisms such as 'GP bashing'). When Sutinen et al explored organisational fairness in hospital doctors they found an association between low organisational fairness and the risk of psychological distress.[19] Dollard and Bakker's theoretical model of Psychosocial Safety Climate highlights the need for policies, practices and procedures to protect workers' psychological health and safety.[20] Our interviewees identified organisational and cultural elements that were causing fear and anxiety and leading them to feel 'unsafe' in their role. Where perceived psychosocial safety is low, workers may experience long-term, high job demands and increased pressure to hide emotions, especially if worker concerns are not listened to.[20] Not being listened to and stigma related to seeking support, were described by interviewees in our study. GPs in the current study talked about different routes they had taken to try and balance their personal resources with the demands of their role and, consequently, make their role in direct patient care more sustainable. However, the effectiveness and viability of these 'coping strategies' were not necessarily long-term. This could lead to increased risk of burnout, identified by Orton et al as prevalent in the UK GPs.[21] Emotional exhaustion (a signal of the development of burnout) can be common among GPs and is associated with older age, high workload, fear of medical errors and

feelings of isolation at work[22] (all factors described by GPs in the current study).

While the themes identified are not unique to GPs, they can be considered fundamental to the satisfaction and sustainability of the workforce and consequently need to be addressed. With 90% of all patient contacts taking place in primary care,[23] a failure to adequately address the GP workforce crisis will have profound ramifications across the NHS. Creating a fairer, safer and more supportive work environment will be fundamental to aiding retention and policies and strategies need to account for this.

### Strengths and weaknesses

The number of interviews conducted provided rich data and the opportunity to explore opinions and experiences with a range of GPs and stakeholders. A maximum variation sample was possible through use of the workforce census survey returns. Identification of stakeholders enabled us to approach participants across South West England, with a range of roles within key organisations.

PPI and project team discussion enabled modification of the original sample targets to ensure that we also captured the views and experiences of 'staying' GPs. The PPI group and GP representative, along with reflective practice incorporating interview field notes and researchers' memos, supported the analysis process.

GPs were self-selecting and represented mainly those indicating the intention to leave (or who had already left) direct patient care. We acknowledge the possibility that those GPs who did not respond to the survey or who were not available for interview may have reported different experiences. The vast majority of GPs eligible for interview were white; GPs from other ethnic groups were approached but none agreed to be interviewed.

### Patient and public involvement group discussion of findings

The group expressed understanding of the pressures that GPs can experience and noted the potential negative impact on patients when GPs were under pressure. There was agreement that more involvement and inclusion of patient participation groups (PPGs) could benefit GPs: positive interactions with patient representatives could help to reduce GP anxiety (eg, around fear of complaints). The PPI members also felt that there was a role for PPGs to support GPs: addressing patient demands and expectations and helping GPs to feel more valued. For PPGs to be of value to practices, it was noted that there is a need for practice staff and patient representatives to be perceived as 'on the same side' and for GPs and other non-clinical staff to trust PPG representatives as part of the practice team. This, in turn, could help the identification of models and examples of good practice that could then be shared by PPGs with other practices.

### CONCLUSION

There is a need to address GP retention in ways that take into account the lived experiences of GPs. The solutions to the present crisis in GP workforce capacity do not lie in short-term initiatives or attempts to boost GP resilience. Effective strategies will need to demonstrate understanding of the key role and value of general practice, to manage the risks inherent in providing general practice and to provide a range of viable ways in which GPs can continue to contribute their key role to the NHS patient care. Showing such commitment to GPs, as central providers of healthcare in the UK, may also prove to be a positive step in attracting new doctors into this clinical specialty.

**Acknowledgements** The authors are grateful to all of the GP and stakeholder participants who contributed interviews, to the PPI group chaired by JW who support the wider programme of work of which the qualitative interviews are a part, to AA, patient representative, to LS, GP representative and to other members of the study advisory group.

**Contributors** AS and RT contributed to the study design, data collection, analysis and writing of the paper. JLC and SGD contributed to study design, analysis and writing of the paper. SR, CS, AA, LS, EF, LL and JW contributed to the study design and writing of the paper. All authors read and approved the final paper. JLC is the guarantor of the paper.

**Funding** The project was funded by the National Institute for Health Research, Health Service and Delivery Research programme (project 14/196/02). The views and opinions expressed herein are those of the authors and do not necessarily reflect those of the Health Service and Delivery Research programme, the National Institute for Health Research, the National Health Service or the Department of Health. JW is generously supported by a Wellcome Trust Institutional Strategic Support Award (WT105618MA). SD's position is partially supported by the National Institute for Health Research (NIHR) Collaboration for Leadership in Applied Health Research and Care South West Peninsula at the Royal Devon and Exeter NHS Foundation Trust.

**Competing interests** All authors have completed the ICMJE uniform disclosure at www.icmje.org/coi_disclosure.pdf and declare: financial support for the submitted work was received from the National Institute for Health Research (HS&DR); JW is supported by a Wellcome Trust Institutional Strategic Support Award; AA has received personal fees from Northern Eastern Western Devon CCG, Devon Local Medical Committee, British Medical Association, University of Exeter, SouthWest CLAHRC and NHS England Medical Directorate (South), outside of this work.

**Ethics approval** University of Exeter Medical School Research Ethics Committee (UEMS REC reference 15/11/085, 3 December 2015).

**Provenance and peer review** Not commissioned; externally peer reviewed.

**Data sharing statement** No additional data are available.

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
