## [Reviewer comments · BMJ Open]

ARTICLE DETAILS

TITLE (PROVISIONAL)	Why do GPs leave direct patient care and what might help to retain them? A qualitative study of GPs in South West England.
AUTHORS	Sansom, Anna; Terry, Rohini; Fletcher, Emily; Salisbury, Chris; Long, Linda; Richards, Suzanne; Aylward, Alex; Welsman, Jo; Sims, Laura; Campbell, John; Dean, Sarah

VERSION 1 – REVIEW

REVIEWER	Dr Sharon Spooner University of Manchester UK shared a workshop and preparation of a short report with 2 of the co-authors of this paper. I have not otherwise worked with them and do not believe any conflict of interest applies in this review.
REVIEW RETURNED	16-Oct-2017

GENERAL COMMENTS	I think this paper adds significant fresh insights on the very topical and serious issue of retention of GPs in the UK and is worthy of publication - though I believe a few revisions should be considered. 1. The Abstract states an objective ('impact of these decisions') which is not reflected in the paper2. In the Results section of the Abstract - 'increased understanding of the lived experience of being a GP in today's NHS-' would benefit from clarification that the results report on a specific sub-group of GPs.3. In the Conclusion - 'Future policies and strategies aimed at retaining GPs in direct patient care should clarify GPs' role and identity...' I feel the paper is not really about clarifying a comprehensive range GPs' roles and identities- rather it presents evidence which support an argument that, to enhance retention of the GP workforce, future policies and strategies should take account of how these are aligned with GPs' sense of roles and identities (this statement is repeated in the Conclusion of the paper).4. Re 'The analysis process was supported through study input from the Patient and Public Involvement group and GP representative.' I remain uncertain of the roles played by PPI/stakeholders in the analysis process. I feel this needs redraft or a better explanation of what is meant by this point. There is also a need for greater detail about what was achieved by the involvement of stakeholders, i.e. how their contributions add to understanding the factors influencing GPs' decisions.
---

	5. Analysis seems somewhat under-described making it difficult to replicate the study. 6. Results - 'In-depth analysis of interview data revealed three themes underpinning GPs' thinking and rationale in respect of continuing to provide direct patient care: ...' I believe there is room for acknowledging the (likely) existence of additional themes which are not reported in this paper. 7. I do not feel that the combination of Fear and Risk works well as one sub-theme; data presented focuses on GPs' concerns about being able to practice safely and their risk of legal action rather than the risks to patient care - as seems implied early in this sub-section. 8. In Strengths and weaknesses - two aspects are mentioned here but barely/not mentioned in the text - local or regional perspectives - was there clear variation/consistency? - mention is made of field notes - but nothing in the manuscript to confirm how/whether these added to understand the experiences of GPs 9. I cannot see an attached checklist (e.g. COREQ)
--	--

REVIEWER	Dr Rebecca Fisher The Health Foundation, UK
REVIEW RETURNED	25-Oct-2017

GENERAL COMMENTS	This is a well conducted study addressing an important problem. The crisis in recruitment and retention in general practice is well known, and early figures suggest that the strategies implemented as a result of the General Practice Forward View are not achieving desired progress towards the target of an additional 5000 GPs by 2020. This paper aims to shed light on the decision making processes around staying in direct patient care. It has clear and timely policy relevance. In my view the methods used are robust. The authors acknowledge that the sample does not include any BAME GPs, and this is a limitation. It would be helpful to know whether their interpretation of 'the South West' includes any major urban centres (e.g Bristol)? This is likely to affect generalisability. The stated objective of the study is to identify factors influencing GPs' decisions about whether or not to remain in direct patient care in general practice, and what might help to retain them in that role. In my opinion the second part of that objective that is the more valuable contribution this study could make. Several other studies have explored the current pressures on GPs, but to my knowledge there is little evidence base to explore GP perceptions of what would help retention. I was therefore slightly disappointed that much of the presented analysis focussed on describing the challenges faced by GPs, rather than exploring what they felt would encourage them to remain in direct patient care. If the interviews explored this then expanding upon it in the paper would be valuable.
---

	Such analysis might help make this paper more relevant to policy makers. At present the conclusions presented are rather vague (particularly in the 'what this study adds' section; e.g. ' to provide a range of viable ways in which GPs' can contribute to the workforce'). Additional clarity, should the authors be able to provide it, might enable this paper to make a more significant contribution to improving policy.
--	---

VERSION 1 – AUTHOR RESPONSE

Reviewer 1: Comments

I think this paper adds significant fresh insights on the very topical and serious issue of retention of GPs in the UK and is worthy of publication - though I believe a few revisions should be considered.

Thank you for your comments. We appreciate this opportunity to consider your suggestions.

Comment: 1. The Abstract states an objective ('impact of these decisions') which is not reflected in the paper

Response: Thank you for spotting this incongruence. We are happy to remove this objective from the paper.

Changes to paper (and page number): "impact of these decisions" removed from Abstract (p.2)

Comment: 2. In the Results section of the Abstract - 'increased understanding of the lived experience of being a GP in today's NHS-' would benefit from clarification that the results report on a specific sub-group of GPs.

Response: 2. In the Results section of the Abstract - 'increased understanding of the lived experience of being a GP in today's NHS-' would benefit from clarification that the results report on a specific sub-group of GPs.

Changes to paper (and page number): "Three key themes underpinned the interviewed GPs' thinking and rationale..."

These themes provide increased understanding of the lived experiences of working in today's NHS for this group of GPs." (p.2)

Comment: 3. In the Conclusion - 'Future policies and strategies aimed at retaining GPs in direct patient care should clarify GPs' role and identity...' I feel the paper is not really about clarifying a comprehensive range GPs' roles and identities- rather it presents evidence which support an argument that, to enhance retention of the GP workforce, future policies and strategies should take account of how these are aligned with GPs' sense of roles and identities (this statement is repeated in the Conclusion of the paper).

Response: 3. In the Conclusion - 'Future policies and strategies aimed at retaining GPs in direct patient care should clarify GPs' role and identity...' I feel the paper is not really about clarifying a comprehensive range GPs' roles and identities- rather it presents evidence which support an argument that, to enhance retention of the GP workforce, future policies and strategies should take account of how these are aligned with GPs' sense of roles and identities (this statement is repeated in the Conclusion of the paper).

Changes to paper (and page number): 3. In the Conclusion - 'Future policies and strategies aimed at retaining GPs in direct patient care should clarify GPs' role and identity...' I feel the paper is not really about clarifying a comprehensive range GPs' roles and identities- rather it presents evidence which support an argument that, to enhance retention of the GP workforce, future policies and strategies should take account of how these are aligned with GPs' sense of roles and identities (this statement is repeated in the Conclusion of the paper).

Comment: 4. Re 'The analysis process was supported through study input from the Patient and Public Involvement group and GP representative.' I remain uncertain of the roles played by PPI/stakeholders in the analysis process.
I feel this needs redraft or a better explanation of what is meant by this point.

There is also a need for greater detail about what was achieved by the involvement of stakeholders, i.e. how their contributions add to understanding the factors influencing GPs' decisions.

Response: The statement quoted appears in the Strengths and Limitations summary bullet points. There is further expansion of the PPI group role in the Public and Patient Involvement section on p.4. We have left the bullet point and added further detail to the PPI section. We hope this now provides sufficient detail.

We agree that we had not specified much detail about what was achieved by the involvement of stakeholders. We have clarified that stakeholders were highly involved in the study as participants, and that 11 of the 19 stakeholders were also GPs – thus able to give dual perspectives.

Changes to paper (and page number): “They also took part in group discussions with the researchers to help refine themes and interpret the data during the analysis process; and contributed to discussion about the implications of the findings.” (p.5)
“Stakeholders provided greater perspectives of the broader issues; 11 of the stakeholders had also been, or were currently, GPs (in addition to any other role).” (p.5)

Comment: 5. Analysis seems somewhat under-described making it difficult to replicate the study.

Response: Thank you for this opportunity to provide more detail. We have added an additional paragraph to describe the analysis process that took place.

Changes to paper (and page number): “Transcripts from GP and stakeholder interviews were analysed together. Transcribed interviews were entered into data management software QSR NVivo11 [13] and analysed using thematic analysis. An initial coding frame was independently constructed by AS and RT, based on five transcripts. Following discussions, a consensus about the coding frame was reached. The coding frame was then independently tested by AS and RT with two further interview transcripts, and final modifications were made. All transcripts were coded using this final coding frame. Detailed project notes were kept regarding further refinement of any existing, or the addition of new, codes. To aid trustworthiness and reduce any potential bias, the researchers wrote field notes and reflexive memos and discussed these during peer debriefing sessions.

Discussions to help analyse emerging themes were held by the research team along with the PPI group and a GP representative (who was not a study participant).”(p.4)

Comment: 6. Results - 'In-depth analysis of interview data revealed three themes underpinning GPs' thinking and rationale in respect of continuing to provide direct patient care: ..'

I believe there is room for acknowledging the (likely) existence of additional themes which are not reported in this paper.

Response: Thank you for highlighting this and for giving us an opportunity to mention the ReGROUP study report to NIHR. Additional themes are discussed in our full report to the funder: NIHR, HSDR study number 14/196/02.

Changes to paper (and page number): The full study report is now cited in the paper.

“These themes are presented in this paper. Additional findings from the interviews, broader than the scope of this paper, are provided in our full ReGROUP study report to the funder (NIHR study number 14/196/02).” (p.6)

Comment: 7. I do not feel that the combination of Fear and Risk works well as one sub-theme; data presented focuses on GPs' concerns about being able to practice safely and their risk of legal action rather than the risks to patient care - as seems implied early in this sub-section.

Response: Thank you for raising this. The research team had a lot of discussion about this theme and recognise that it is a challenging one to present. We have provided additional data to support the inclusion of our statement around GPs' concerns about patient safety. We wish this sub-theme to remain and hope that the additional data supports this.

Changes to paper (and page number): Additional text and participant quote added to p.8

“There were concerns about the safety of practice and the quality of care being delivered to patients: “...when repeat prescriptions came through to be re-authorised I would be checking through and making sure everything was up to date and everybody else (...) was just re-authorising it because they'd given up that aspect (...) of safety...” (GP locum, female, age 40-49, staying)
Risk was related to ‘unmanageable’ workloads...”

Comment: 8. In Strengths and weaknesses - two aspects are mentioned here but barely/not mentioned in the text

a) local or regional perspectives - was there clear variation/consistency?

b) mention is made of field notes - but nothing in the manuscript to confirm how/whether these added to understand the experiences of GPs

Response: Thank you.

a) We agree that the main text does not describe variation/consistencies within the local and regional perspectives. This line has now been removed from the text.

b) This is addressed in the expanded analysis description (see point 5 above)

Changes to paper (and page number): Reference to local and regional perspectives has been removed from this section of the paper.

Reviewer 2: Comments

Comment: This is a well conducted study addressing an important problem. The crisis in recruitment and retention in general practice is well known, and early figures suggest that the strategies implemented as a result of the General Practice Forward View are not achieving desired progress towards the target of an additional 5000 GPs by 2020. This paper aims to shed light on the decision making processes around staying in direct patient care. It has clear and timely policy relevance.

Response: Thank you for your comments and for acknowledging the clear and timely policy relevance of this paper.

Comment: In my view the methods used are robust. The authors acknowledge that the sample does not include any BAME GPs, and this is a limitation. It would be helpful to know whether their interpretation of 'the South West' includes any major urban centres (e.g Bristol)? This is likely to affect generalisability.

Response: Thank you for highlighting this. We have now included a statement in the Results section that notes the inclusion of participants from urban and rural areas within the South West.

Changes to paper (and page number): Text added:

“Interviewees were recruited from urban and rural areas across South West England.” (p.5)

Comment: The stated objective of the study is to identify factors influencing GPs' decisions about whether or not to remain in direct patient care in general practice, and what might help to retain them in that role. In my opinion the second part of that objective that is the more valuable contribution this study could make. Several other studies have explored the current pressures on GPs, but to my knowledge there is little evidence base to explore GP perceptions of what would help retention. I was therefore slightly disappointed that much of the presented analysis focussed on describing the challenges faced by GPs, rather than exploring what they felt would encourage them to remain in direct patient care. If the interviews explored this then expanding upon it in the paper would be valuable.

Such analysis might help make this paper more relevant to policy makers. At present the conclusions presented are rather vague (particularly in the 'what this study adds' section; e.g. 'to provide a range of viable ways in which GPs' can contribute to the workforce'). Additional clarity, should the authors be able to provide it, might enable this paper to make a more significant contribution to improving policy.

Response: Thank you for drawing our attention to this. The study presented in this paper is one part of a comprehensive programme of work, seeking to identify implementable policies and strategies to support the retention of experienced GPs in direct patient care and to support the return of GPs following a career break. We have added reference to the reporting of this entire programme of work in the results section (p.6)

The key finding from this qualitative study was the insight into the 'lived experience' of being a GP in today's NHS and how this can impact on decisions about whether to remain in, or to leave, direct patient care. We feel that this perspective, and the three key themes arising, adds to the current literature around GP workforce retention. Through a discussion of workplace theory, we suggest that future policies and strategies should take account of GPs' lived experiences in order to support and encourage them to remain in direct patient care.

We have added the second bullet point from the 'what this study adds' section to the Conclusion. We hope that this, plus the expanded explanation here, is sufficient to address this comment.

The 'what this study adds' section has been removed on request of the editorial board.

Changes to paper (and page number): Thank you for drawing our attention to this. The study presented in this paper is one part of a comprehensive programme of work, seeking to identify implementable policies and strategies to support the retention of experienced GPs in direct patient care and to support the return of GPs following a career break. We have added reference to the reporting of this entire programme of work in the results section (p.6)

The key finding from this qualitative study was the insight into the 'lived experience' of being a GP in today's NHS and how this can impact on decisions about whether to remain in, or to leave, direct patient care. We feel that this perspective, and the three key themes arising, adds to the current literature around GP workforce retention. Through a discussion of workplace theory, we suggest that future policies and strategies should take account of GPs' lived experiences in order to support and encourage them to remain in direct patient care.

We have added the second bullet point from the 'what this study adds' section to the Conclusion. We hope that this, plus the expanded explanation here, is sufficient to address this comment.

The 'what this study adds' section has been removed on request of the editorial board.

VERSION 2 – REVIEW

REVIEWER	Dr Sharon Spooner University of Manchester, UK
REVIEW RETURNED	01-Dec-2017

GENERAL COMMENTS	I am very happy with the revisions made following initial reviews and to recommend publication of this paper.
---

REVIEWER	Rebecca Fisher The Health Foundation, UK
REVIEW RETURNED	21-Nov-2017

GENERAL COMMENTS	I am satisfied that the authors have addressed concerns raised in the first reviews. This paper makes an important and timely contribution, and in my view is suitable for publication.
---